# The Deficits of Insulin Signal in Alzheimer's Disease and the Mechanisms of Vanadium Compounds in Curing AD

Jinyi Yao [1], Zhijun He [1], Guanying You [1], Qiong Liu [1] and Nan Li [1,2,*]

[1] Shenzhen Key Laboratory of Marine Biotechnology and Ecology, College of Life Sciences and Oceanography, Shenzhen University, Shenzhen 518055, China; 2100251036@email.szu.edu.cn (J.Y.)
[2] Shenzhen Bay Laboratory, Shenzhen 518055, China
\* Correspondence: lin@szu.edu.cn; Tel.: +86-13715189671; Fax: +86-75526536629

**Abstract:** Vanadium is a well-known essential trace element, which usually exists in oxidation states in the form of a vanadate cation intracellularly. The pharmacological study of vanadium began with the discovery of its unexpected inhibitory effect on ATPase. Thereafter, its protective effects on β cells and its ability in glucose metabolism regulation were observed from the vanadium compound, leading to the application of vanadium compounds in clinical trials for curing diabetes. Alzheimer's disease (AD) is the most common dementia disease in elderly people. However, there are still no efficient agents for treating AD safely to date. This is mainly because of the complexity of the pathology, which is characterized by senile plaques composed of the amyloid-beta (Aβ) protein in the parenchyma of the brain and the neurofibrillary tangles (NFTs), which are derived from the hyperphosphorylated tau protein in the neurocyte, along with mitochondrial damage, and eventually the central nervous system (CNS) atrophy. AD was also illustrated as type-3 diabetes because of the observations of insulin deficiency and the high level of glucose in cerebrospinal fluid (CSF), as well as the impaired insulin signaling in the brain. In this review, we summarize the advances in applicating the vanadium compound to AD treatment in experimental research and point out the limitations of the current study using vanadium compounds in AD treatment. We hope this will help future studies in this field.

**Keywords:** vanadium; Alzheimer's disease; diabetes; insulin resistance; mitochondrial; oxidative phosphorylation



## 1. Introduction

Alzheimer's disease (AD) is the most prevalent dementia disease in aging people. The histopathological definitions of AD are the presence and the deposition of amyloid-beta (Aβ) extracellularly and the accumulation of tau-neurofibrillary tangles (NFT) intracellularly, plus severe neurodegeneration along with impaired cognitive abilities [1]. The aggregation of tau is not only present in AD but is also shown in specific subtypes of frontal temporal dementia (FTD). On the other hand, the mutation amyloid precursor protein (APP) and γ-secretase, which lead to the over-production of Aβ, were found to trigger the onset of familial AD. Interestingly, tau and APP are all corelated with glucose metabolism and insulin signaling pathways, as discussed followed.

In this article, we review the relationship between tau, APP, and glucose metabolism for the insulin signal. We also evaluate the protective effects of vanadium compounds, the anti-diabetes agents, on AD treatment and discuss the underlying mechanisms of these agents, as well as the limitation of current studies in this field by searching for keywords in pairs, including APP and glucose/insulin, tau and glucose/insulin, vanadium and glucose/insulin, Alzheimer's disease and vanadium on Pubmed.

## 2. The Deficits of Insulin Signal in AD

### 2.1. The Role of APP in Glucose Metabolism

The extracellular deposits of amyloid-beta (Aβ) plaques and the intracellular neurofibrillary tangles (NFT) formed by hyperphosphorylated tau are well-known histopathological characters of AD, which are accompanied by damaged mitochondria in the neuron and the severe atrophy of the central nervous system (CNS).

It seems that the level of Aβ correlated to onset of AD, as indicated by familial AD (FAD) patients, including those carrying mutations on the amyloid precursor protein (APP) [2], a disintegrin and metalloproteinase 10 (ADAM10, an α-secretase) [3,4] and/or presenilin-1/2 (PS1/2, the components of γ-secretase) [5] which give rise to the overproduction of Aβ, probably suffer from AD during their younger life. The typical neuropathology of AD seen in Down syndrome also emphasizes the toxicant of Aβ. The gene encoding APP is located on chromosome 21. The trisomy of 21 patients harboring three copies of APP exhibited abundant diffused Aβ plaques in their CNS and invariably obtained AD pathologies after a certain age. Aβ is produced by the cleavage of APP, which is a type 1 transmembrane protein, by β-secretase (BACE1) and γ-secretase to generate three fragments, including a soluble APPβ ectodomain, an Aβ domain, and an APP intracellular domain (AICD). However, when APP is hydrolyzed by α-secretase, it generates the APPα ectodomain, which is longer than APPβ ectodomain, without producing the Aβ fragment. Thus, this is not an amyloidogenic process.

The toxicities of Aβ have been intensively studied. It was observed that Aβ oligomers could assemble to form pores on cell membranes for ion transportation and impair the appropriate permeability of membranes [6], which resulted in the depolarization of microglia and neuron [7]. Soluble Aβ could also impair synaptic plasticity through the over-activating NMDA receptor [8], which resulted in mitochondria damage [9,10]. In addition, Aβ oligomers were demonstrated to induce inflammatory reactions through a Toll-like receptor [11] and perturb the blood–brain barrier [12]. Although Aβ overproduction is considered the most pivotal risk factor for AD development, it was observed that many elderly non-dementia people also carried Aβ plaques in their brains. Recently, it has been confirmed that the severity of dementia is dependent on the NFT burden but not the level of Aβ senile depositions [13]. Therefore, many scientists suggested that AD should be considered as a secondary tau pathology. This idea is also supported by the discovery that two people carrying the PS1-E280A mutation, which usually results in typical AD before they are 50 years old, did not develop dementia before age 70. They all had severe Aβ plaque burdens in their brains, but they did not develop tau pathology in their brains as other PS1-E280A mutation carriers did. One of them included a APOE3-R136S homozygote [14], and the other one a RELN-H3447R mutation carrier [15].

Is the function of APP aimed to produce Aβ, which is a toxicant for the brain? The answer must be no. It has been found that APP plays an important role in glucometabolic. For example, App knockout mice reduced plasma glucose compared to the wild-types (WT) [16]. When mice were treated with glucose or a membrane-permeant cAMP, insulin secretion in *App* knockout mice increased much higher than that in WT [17]. More interestingly, the APP deficiency resulted in mice being resistant to diet-induced obesity and having higher energy expenditure at night [18]. Meanwhile, the level of insulin was lower in the brains of *App*-ablated mice because of the increase in the insulin-degrading enzyme (IDE), and the synaptosomes prepared from *App*-ablated mice showed diminished insulin receptor phosphorylation compared with WT mice [19]. On the other hand, the APPα fragment of APP, which was generated by α-secretase hydrolyzation, also modified the phosphorylation of Akt [20], indicating that APP itself is involved in glucometabolic.

### 2.2. The Influence of Tau on Insulin Signal

Despite the terrible toxicity of Aβ seen in vivo and in vitro, a great many older people bearing Aβ plaques in their brains have not exhibited dementia symptoms until the tau pathology has appeared [21]. This may be due to the sequestration of Aβ plaques by

microglia [22]. Tau is a microtubule-associated protein that was believed to stabilize microtubules and facilitate cargo transport. It is encoded by *MAPT* on chromosome 17. In the human brain, the exons 2, 3, and 10 of *MAPT* can be alternatively spliced, the former two encoding two N-terminal repeats (N), with the latter one encoding a microtubule-binding repeat (R) domain. There are four microtubule-binding repeats in total. Therefore, the alternative splicing of *MAPT* produces six distinct tau isoforms, which are 0N3R, 1N3R, 2N3R, 0N4R, 1N4R, and 2N4R. All of them could be detected in the paired helical filaments of AD.

A great many efforts have been made to disclose how Aβ can trigger tau pathology; thus, the conventional Aβ cascade hypothesis of AD pathophysiology can be integrated. It was found that the Aβ oligomer activated Fyn through the prion protein (PrP) [23], leading to the hyperphosphorylation of tau [24]. It was also demonstrated that oligomeric Aβ overstimulated the N-methyl-D-aspartate receptor (NMDAR), which, in turn, triggered cyclin-dependent kinases 5 (CDK5) activation and tau phosphorylation [25] (Figure 1). In addition, it was shown that Aβ was able to attenuate insulin signaling and activate glycogen synthase kinase -3 (GSK-3β), which resulted in tau phosphorylation [26]. Moreover, Aβ was found to increase tau proteolysis at Asp421 and exacerbate the rate and extent of the tau filament assembly in vitro [27].

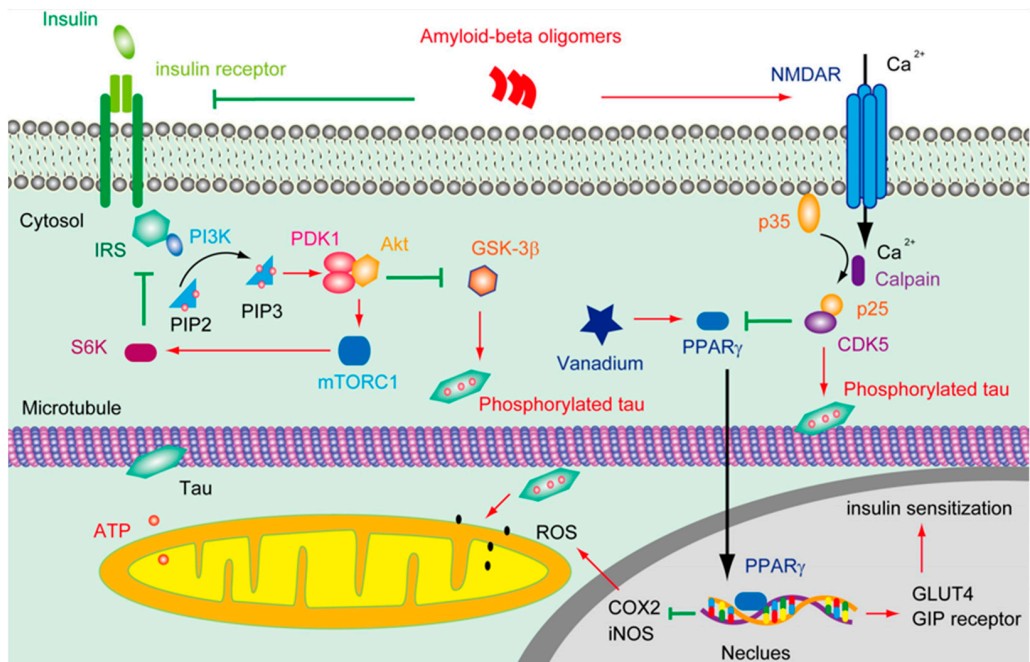

**Figure 1.** The deductive effluence of vanadium compounds on insulin signal in AD pathology.

Notably, there are many other tau pathologies besides AD, such as Pick's disease (PiD), chronic traumatic encephalopathy (CTE), argyrophilic grain disease (AGD), corticobasal degeneration (CBD), progressive supranuclear palsy (PSP), and a subclass of frontotemporal dementia with Parkinsonism linked to chromosome 17 (FTDP-17tau) [28]. Tau is hyperphosphorylated not only in the NFT of AD but also in other tau pathologies. There are many kinases that are involved in tau hyperphosphorylation, including the death-associated protein kinase 1 (DAPK) which is also associated with the late-onset of AD [29,30], $Ca^{2+}$/Calmodulin-dependent protein kinase II (CAMKII) which is involved in LTP formation [31], and Fyn, CDK5, GSK-3β as aforementioned above. On the other hand, the dysfunction of protein phosphatase 2A (PP2A) was also shown to be responsible for the intensive phosphorylation of tau [32]. The hyperphosphorylation of tau resulted in the dissociation of tau and microtubules [33,34]. However, it seems that the NFT itself was not sufficient to cause cognitive decline or neuronal death alone [35].

The acetylated tau was also seen in tauopathies due to the dysregulation of both p300 acetyltransferase and sirtuin 1 (SIRT1) deacetylase [36]. The acetylation of tau inhibited the chaperone-mediated clearance of tau and promoted tau propagation in mice [37]. The inhibition of p-300-induced tau acetylation by salsalate reduced the tau level and prevented hippocampal atrophy [38]. Attractively, it was found that the acetylation of tau was significantly enhanced in high-glucose-treated cells. By contrast, the activation of AMP-activated protein kinase (AMPK) ameliorated the acetylation of tau and rescue memory impairments in a SIRT1-dependent manner in the mice model [39]. AMPK is involved in glucose metabolism. Upon activation by liver kinase B1 (LKB1), transforming the growth factor β-activated kinase 1 (TAK1), AMPK could regulate the level of peroxisome proliferator-activated receptor gamma coactivator 1 α (PGC-1α) [40].

Before asking about the neurotoxicity of tau phosphorylation and/or aggregation, one may want to know the basic functions of tau itself. Indeed, except for binding to the microtubule, tau is involved in regulating insulin signaling as well. It was found that tau interacted with the tension homolog on chromosome 10 (PTEN) and exerted an inhibitory effect on its lipid phosphatase activity. The knockout of tau resulted in the activation of PTEN, and the dephosphorylation of PtdIns(3,4,5), thus impairing the hippocampal's response to insulin-induced LTD in brain slices [41]. It was also reported that tau ablation in mice leads to pancreatic β cell dysfunction and glucose intolerance [42]. In addition, tau knockdown increased the basal insulin level but perturbed glucose-stimulated insulin secretion [43]. Interestingly, it was also observed that the phosphorylation of tau resulted in the intraneuronal accumulation of insulin oligomers and insulin signaling deficits [44]. However, in the streptozotocin (STZ)-induced type 1 diabetes model mice, tau knockout attenuated the cognitive impairment triggered by insulin deficiency [45], whereas, in the same conditions, human tau transgenic mice showed severe impairments in learning and memory [46]. In addition, in P301L mutation knock-in male mice but not female mice, a high-fat diet triggered higher insulinemia and glucose intolerance compared with wild-type littermates [47]. These studies suggested that tau is closely correlated with insulin signaling and glucometabolic instead of only participating in microtubule stabilization.

Aβ overproduction resulted in $Ca^{2+}$ influx through NMDAR, which, in turn, activated CDK5 via the calpain mediate cleavage of p35 into p25. CDK5 subsequently phosphorated tau and suppressed the activity of PPARγ. Tau phosphorylation and truncation impair the functions of the mitochondria and increase the level of ROS. However, vanadium can activate PPARγ, which is involved in facilitating insulin secretion and maintaining insulin receptor activation through upregulating the GIP receptor and IRS, thus restraining the hyperphosphorylation of tau; on the other hand, the activation of PPARγ by vanadium may protect the mitochondria from the accumulation of ROS by downregulating the level of cyclooxygenase-2 (COX2) and inducible nitric oxide synthase (iNOS).

### 2.3. The Impaired Insulin Signal in AD

"Type 3 diabetes" was first used to describe AD by Steen, E. et al. [48] for the abnormal levels of insulin and glucose in CSF [49], as well as the insulin resistance that was found in the brains of AD patients [50]. Type 3 diabetes is not a medical-approved term, however, and it has been demonstrated that the Ab oligomer interrupts the activation of PI3K and abolishes the suppression of insulin on GSK-3β, which is involved in triggering the hyperphosphorylation of tau besides energy metabolism [51]. In addition, the IDE is able to decompose both insulin and Aβ [52]. In IDE-deficient mice, the level of endogenous soluble Aβ was elevated in the brain. On the contrary, the overexpression of IDE in the neuron of APP transgenic mice significantly reduced the level of soluble Aβ and postponed the formation of amyloid plaque. Interestingly, in the brain of those who carry apolipoprotein E-epsilon 4 (APOE4), the most significant genetic risk factor for sporadic AD, the protein level of IDE, was reduced by approximately 50% [53]. However, in the blood–brain barrier of AD with cerebral amyloid angiopathy (CAA), the level of IDE was enhanced [54], which could impair the transportation of insulin from the periphery to the CNS. Moreover,

when insulin was depleted in mice, both tau phosphorylation and tau filaments were reinforced in the brain [55]. In line with these observations, depleting insulin by STZ also triggered tau phosphorylation and NFT formation [56]. Moreover, when insulin receptor substrate 2 (IRS2) was lost, the phosphorylation of tau was promoted [57]. Taken together, this evidence coincidentally demonstrates that insulin, the signaling pathway's dysfunction plays a pivotal role between Aβ overproduction and tau pathology.

In brains, insulin can either be derived from in situ de novo synthesis [58] or from the peripheral plasma. Insulin can pass across the capillary endothelial cells of BBB in saturable, selective, and receptor-dependent manners [59,60]. Through stimulating the insulin receptor (IR) and/or insulin-like growth factor 1 receptor (IGF1R), insulin facilitated the phosphorylation of an insulin receptor substrate (IRS) and subsequently activated PI3K and AKT (Figure 1). As a result, the glucose transporter 4 (GLUT4) in the cytosol was translocated onto the plasma membrane to enhance glucose uptake [61]. Insulin triggering the translocation of GLUT4 is very critical in the process of hippocampal-dependent memory consolidation [62]. Of note, the insulin signaling pathway was regulated by negative feedback. Except for stimulating Rho GTPase to facilitate the transportation of GLUT4, the activation of Akt also induced the functioning of a mammalian target of rapamycin complex 1 (mTORC1), which is sensitive to Rapamycin. mTORC1 can further stimulate ribosomal protein S6 kinase (S6K), which inhibits the activity of IRS1, thus silencing the insulin-PI3K-Akt signal. mTORC1 is also involved in regulating some other cellular processes, including autophagy and mitochondrial oxidative respiration. Upon binding to its receptor, insulin can also trigger the activation of growth factor receptor-bound protein 2 (Grb2), which further stimulates Ras, Raf, and mitogen-activated protein kinases (MAPK) [63]. Notably, the hyperactivation of mTORC1 was spotted in the early to mid-stage of AD brains [64]. In terms of MAPK, except for being stimulated by the insulin signal, the overreaction of p38 was also implicated in Aβ induced toxicity [65].

The dysregulation of the insulin signal was also seen in the APOE4 carrier; it was found that the insulin receptor was trapped in the endosomes of primary neurons treated by APOE4 [66]. In addition, the knockout of the triggering receptor expressed on myeloid cell 2 (TREM2), which is a great genetic risk factor following APOE4, also exacerbated insulin resistance [67]. Interestingly, insulin resistance upregulated the expression of GCN5, a histone acetyltransferase, which resulted in an increase in CDK5 and tau phosphorylation [68]. These data indicate that the Aβ overproduction and genetic risk factors of AD can, directly and indirectly, impair the insulin signal, therefore triggering tau phosphorylation. On the other hand, the dysfunction of tau may further induce insulin resistance and/or insulin deficiency in the AD brain.

## 3. The Advance of Applying Vanadium in AD Treatment

### 3.1. The Biology Functions of Vanadium

Vanadium is an essential trace element that plays an important role in the metabolism of cholesterol and triglyceride, as well as the oxidation of glucose and the synthesis of glycogen [69]. Vanadium usually exists in the form of vanadate anion ($VO_3^-$) extracellularly and in the form of vanadyl cation ($VO_2^+$) intracellularly in the body, respectively [70]. Since the ATPase inhibitory effects of vanadate were observed by accident [71], it has been well documented that vanadate was similar to phosphate in size and charge, which gave vanadium the ability to irreversibly prohibit the conformational variety of the dephosphorylated enzyme [72]. Later on, vanadyl ions showed insulin-like features in rat adipocytes [73]. Further studies revealed that peroxovanadates inhibited the function of the protein tyrosine phosphatase (PTPase) [74], which was involved in the dephosphorylation of the insulin receptor and arrested insulin signaling. It was also demonstrated that vanadyl bisacetylacetonate exerted an antilipolytic influence via activating Akt (protein kinase B, PKB) [75,76]: a key kinase downstream of the insulin-PI3K (phosphatidylinositol-3-kinase) signaling pathway. Moreover, it was reported that the protein level of PPARγ (peroxisome proliferator-activated receptor gamma), a transcriptional factor that was shown to reduce

insulin resistance [77] upon its activation in β-pancreas cells [78] and adipocytes [79], was modulated by vanadyl bisacetylacetonate.

Though the biological functions of vanadium were well documented, its toxicity was also seen in animal studies [80,81]. The clinical study revealed that the consumption of vanadium at 125 mg/day was safe for adults [82]. However, rats all died when they received vanadyl sulfate for more than 2 mM/kg of their body weight [83]. The biological effects are different according to the species of vanadium compounds [84]; the toxicities of vanadium in different oxidation states were divergent as well. Studies have illustrated that the highest oxidated valence (+5) of vanadium was the most toxic state [85]. This is because, in this state, the strong prooxidant property of vanadium severely aggrandizes oxidative stress [86] and perturbs the mitochondria [87]. In pharmacological studies, many vanadium compounds, such as bis(maltolato) oxovanadium(IV) (BMOV) [88], bis(2-ethyl-3-hydroxy-4-pyronato) oxovanadium (IV) (BEOV) [89], N,N-dimethylphenylenediamine-derivatized nitrilotriacetic acid vanadyl complexes (VO(dmada)) [90], vanadyl complex of p-hydroxyl aminophenol derivative (VOphpada) [91], and graphene quantum dots(GQD)-VO(p-dmada) [92], have been synthesized to improve the affordability and stability of inorganic vanadium salts. It was shown that the oral uptake of BEOV increased the absorbance of vanadium 2–3 times in most tissues compared to $VOSO_4$ [93].

Interestingly, it was reported that the level of vanadium declined in the plasma of Alzheimer's disease (AD) patients [94,95], indicating that this trace element could become involved in AD pathology. Recently, the protective effects of vanadium compounds on AD pathology have been observed in different AD mouse models.

### 3.2. Potential Mechanisms of Vanadium Action in AD Curing

It was shown that the administration of insulin could reduce the ratio of tau-phosphorylated tau-181/Aβ42 in plasma and maintain the volume of AD brains [96]. However, long-term insulin administration probably triggered insulin resistance. By contrast, the intranasal administration allowed insulin to arrive in CNS bypasses through the periphery and prevent the risks associated with hypoglycemia [97]. The study on rats indicated that the intranasal insulin supply was able to improve memory and inhibit inflammation in AD [98]. This clinical study also indicated that the verbal memory of Mild Cognitive Impairment (MCI) and AD patients without APOE4 were improved immediately after the 40 IU intranasal insulin supply without perturbing the plasma levels of insulin and glucose [99]. Another trial showed that 40 IU/day of intranasal insulin administration for 21 days significantly ameliorated the working verbal memory and working visuospatial memory [100]. Nevertheless, these effects were affected by APOE alleles. Insulin administration alleviated insulin resistance only in APOE4 carriers but not in APOE3 or APOE2 carriers [101]. Nonetheless, a recent study reported that intranasal insulin administration exhibited no benefits on cognitive functions in a randomized clinical trial including 289 adults with mild cognitive impairment or AD [102]. However, this study had a profound limitation, which is the device that was used in this study for intranasal insulin administration and has not been tested before. Therefore, further research in this field is still needed to illustrate the effects of insulin on AD curing and the underlying mechanism.

In other studies, the effects of insulin sensitizers, which showed benefits in curing type-2 diabetes, were tested for curing AD on different mice models or clinical trials. Peroxisome proliferator-activated receptor (PPAR-γ) agonists, such as rosiglitazone [103] and pioglitazone [104], showed great benefits on AD pathologies. Six months of rosiglitazone administration for 4 mg/day significantly improved selective attention and delayed the recall of AD patients. In addition, 6 months of pioglitazone supplication for 10–30 mg/day decreased fasting plasma insulin levels of AD patients who also suffered from type 2 diabetes mellitus. Meanwhile, the plasma Aβ levels of these subjects were decreased compared with the AD patients in the control group who received a placebo [103,105]. Another study showed that 24 weeks of rosiglitazone administration at 8 mg/day significantly ameliorated the performance of APOE4-negative AD patients [106]. Nonetheless, a phase 3 trial

demonstrated that rosiglitazone had no effects on the cognitive functions of AD, regardless of the APOE type [107]. In addition, it was found that metformin increased the IDE level in transgenic AD mice [108] and prevented amyloid plaque deposition and memory impairment [109]. An in vitro study also revealed that metformin-induced the dephosphorylation of tau through PP2A [110]. Clinically, the use of metformin showed protective effects on brain volumes in non-demented elderly individuals with diabetes [111]. For mild cognitive impairment or mild dementia due to AD, metformin also improved executive functioning [112].

The vanadium (IV) compound could rescue cholinergic neurons in the medial septum of a bilateral olfactory bulbectomy mouse in a dose-dependent manner. The impaired long-term potentiation (LTP) of these mice was also prevented by bis(1-N-oxide-pyridine-2-thiolato)oxovanadium(IV) (VO-(OPT)) [113]. However, these mechanisms have not been studied deeply. Vanadyl (IV) acetylacetonate (VAC) was found to attenuate neuron loss in APP/PS1 transgenic AD model mice and preserve cognitive functions. It up-regulated the expression of glucose-regulated protein 75 (Grp75), thus suppressing p53-mediated neuronal apoptosis without reducing A$\beta$ plaques in the mice brain. Furthermore, the neuroprotective ability of VAC could be correlated with the activation of PPAR$\gamma$ and AMPK signaling [114]. Another vanadium compound, BEOV, significantly reduced the phosphorylation of tau and inhibited A$\beta$-induced inflammation by inhibiting the NF-$\kappa$B signal both in vitro and in vivo [115]. BEOV also blocked the neurotoxicity induced by endoplasmic reticulum (ER) stress by inhibiting Bip and p-eIF2$\alpha$ [116] and ameliorated spatial learning and memory in AD mouse models [117]. More importantly, we found that the biological benefits of BEOV on AD pathologies were dependent on PPAR$\gamma$ [31,116,118], which resembled the functions of bis (5-hydroxy-4-oxo-4H-pyran-2-hydroxy-benzoatato) oxovanadium (IV) (BSOV) [119].

PPAR$\gamma$ is a member of the nuclear hormone receptor family of ligand-inducible transcription factors, which plays a pivotal role in lipid and glucose homeostasis. The activity of PPAR$\gamma$ could be inhibited by CDK5 and MAPK [120] (Figure 1). It was reported that the activation of PPAR$\gamma$ is involved in upregulating the level of the glucose-dependent insulinotropic polypeptide (GIP) receptor [121], GLUT4, and pyruvate carboxylase [122,123], which are correlated with insulin sensitization. In addition, the activation of PPAR$\gamma$ is able to repress the NF-$\kappa$sB-dependent transcription of iNOS and COX2 [124], which are involved in the generation of ROS. Molecular docking analysis revealed that the binding energy of BEOV with PPAR$\gamma$ was ~8.1 kcal mol$^{-1}$, indicating that BEOV interacted quite well with PPAR$\gamma$ and could be an agonist for PPAR$\gamma$ [116]. Though the vanadium compound showed great protective effects on the transgenic AD model mice, it is still unknown whether these anti-diabetes agents are valid in the later stages of AD, which is featured by severe neurotrophy accompanied by the propagation of prion-liked tau.

## 4. The Potential Mechanisms of Vanadium in Curing AD for Future Study

Tau is localized in the mitochondria in addition to their association with the microtubule [125]. In neurons expressing mutated tau found in FTLD, the hyperphosphorylated tau impaired the function of the mitochondria by breaking down complex I of the electron transport chain [126,127]. In addition, it was demonstrated that hyperphosphorylated tau also promoted mitochondrial fission and a morphological change through interacting with dynamin-related GTPase (Drp1) [128]. Moreover, it was shown that the cleavage of tau promoted the formation of NFTs [129]. Importantly, cleaved tau perturbed the mitochondrial dynamics when the intracellular calcium level was increased by thapsigargin treatment as well [130,131]. On the other hand, the reduction in tau has also been found to protect the neuron from the loss of mitochondrial membrane potential loss [132], excitotoxicity [133], and axonal transport inhibition [134] induced by A$\beta$ [135].

The functions of tau on mitochondrial metabolism and homeostasis attract more and more attention these days. It has been shown that the overexpression of human tau resulted in mitochondrial elongation and accumulation, along with a reduction in

ubiquitination of mitofusion 2 (MFN2) [136]. Moreover, in mutated human tau (P301) transgenic mice [137], 3× transgenic AD mice [138], and AD patients [139], the level of MFN2 was reduced [140]. In flies, the overexpression of tau affected the expression of drp1 and Marf (the homologous to human MFN2) [141]. The level of MFN1/2 was reduced in APOE4 carriers [142]. By analyzing genotypes and allele frequencies in the Korean AD population, the rs1042837 polymorphism in MFN2 was involved in the pathogenesis of AD [143,144]. On the other hand, the forced overexpression of MFN2 in P301S human tau transgenic mice suppressed tau pathology-induced neurodegeneration and cognitive decline [145]. It has also been reported that, in tau knockout mice, the protein level of nuclear factor-erythroid-2-related factor 2 (Nrf2) was reduced, while the expression of MFN2 and PGC-1α was significantly increased [146]. MFN2 is a guanosine triphosphatase (GTPase) on the outer membrane of the mitochondria, which is involved in mitochondrial fusion. MFNs form dimers in a GTP-dependent manner to facilitate the membrane tethering ability [147]. MFN1/2 are critical for glucose-stimulated insulin secretion (GSIS) through regulating the mtDNA expression via Tfam [148]. The trafficking of mitochondria-induced by 3,4-methylenedioxymeth-amphetamine (MDMA) is dependent on tau and MFN2/Drp1 [149].

The tau interactome revealed that, except for the microtubule, tau could interact with presynaptic vesicle proteins and mitochondria proteins. More importantly, FTD-related mutations of tau impaired the interaction of tau with mitochondria proteins, including SUCLG1, SUCLG2, SLC25A6, CYCS et al. [150]. In contrast, in the phosphorylated tau interactome that is derived from the NFT of AD, many of these mitochondrial proteins were not found; instead, novel phosphorylated tau interactors were presented, including OXCT1, COX5B, VDAC2, for example [151]. Among these tau-interacting mitochondrial proteins, Oxct1 has been identified as a p–tau interacting protein [151] and a therapeutic target of AD [152], SUCLG2 has been recognized as a determinator of CSF Aβ1-42 levels and [153] and promising for the AD signature protein [154]. Interestingly, these two proteins are involved in a similar biological process, which is the transfer of Co-A from Succinyl-CoA. The difference between them is that SUCLG1/2 catalyzes the only substrate-level phosphorylation in the tricarboxylic acid cycle, and the transfer of CoA is accompanied by the production of GTP in mammals [155]; however, OXCT1 catalyzes the reversible transfer of CoA from succinyl-CoA to acetoacetic acid without the production of GTP [156] (Figure 2).

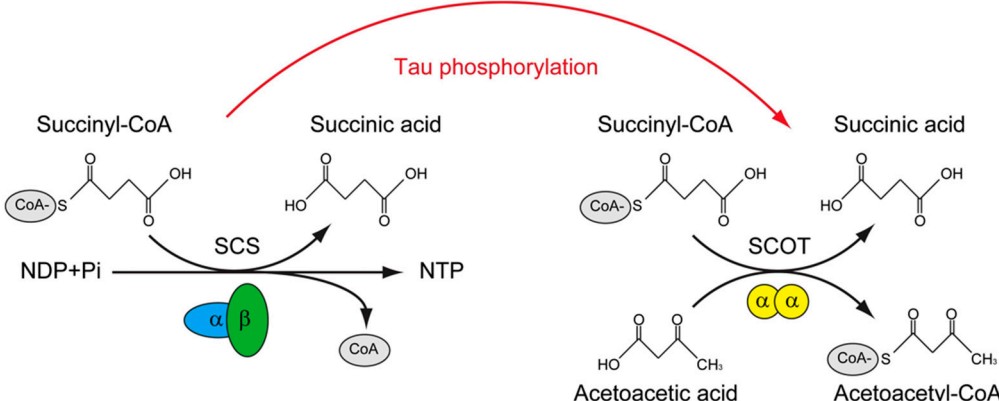

**Figure 2.** The putative role of tau phosphorylation on substrate level phosphorylation of mitochondrial.

Succinyl-CoA synthetase (SCS) catalyzes the only substrate-level phosphorylation in the tricarboxylic acid cycle; the transfer of CoA is accompanied by the production of ATP/GTP. The SCS is a heterodimer, which is composed of SUCLG1 and either SUCLG2A (specific for ATP production) or SUCLG2G (specific for GTP production). The succinyl-CoA: 3-ketoacid-CoA transferase catalyst (SCOT/OXCT1) is a mitochondrial homodimer, which catalyzes the reversible transfer of CoA from succinyl-CoA to acetoacetic acid without the production of ATP/GTP. The levels of ATP/GTP are critical

for mitochondrial-dependent glucose-stimulated insulin secretion. SCS was found to interact with tau, whereas SCOT/OXCT1 was found to interact with phosphorylated tau. Whether tau phosphorylation is involved in regulating the substrate level phosphorylation in mitochondria is of interest to know.

GTP level determines cell fate by regulating Bcl2/Bax expression and the activation of caspase-3 [157] and p53 [158]. Importantly, Bax has been found to positively regulate mitochondrial fusion through MFN2 [159,160]. In addition, Bak was involved in regulating mitochondrial morphology and pathology during apoptosis by interacting with MFNs [161]. Whether tau pathology perturbs the level of GTP is unknown, while the experimental results indicated that the phosphorylation of tau may perturb the substrate level of phosphorylation and mitochondrial-dependent GSIS. This is also evidenced by the observation that tau knockout can prevent the neurotoxicity induced by Aβ peptide [132,135,162] and the stress induced by dendritic atrophy [163] and type 1 diabetes-induced cognitive impairment [45]. In addition, the tau ablation also improved mitochondrial function by increasing the levels of MFN2 and increasing ATP production in the hippocampus [146]. The vanadium compound BEOV was found to significantly inhibit tau phosphorylation at Ser396 and Ser404 in the primary neuron and brain of the transgenic AD mice model and improved the spatial learning activity of these mice [164]. However, whether vanadium had any influence on the mitochondria functions is unknown. In future studies, the function of the mitochondrial need to be recruited into this field.

## 5. Conclusions and Perspectives

Numerous studies have demonstrated that tau is not only localized within the mitochondria [125] but also exerts pivotal functions in mitochondrial metabolisms. Apart from inducing mitochondrial abnormalities by hyperphosphorylated tau [140], the caspase 3-cleaved tau also impaired mitochondrial dynamics in AD [131,165]. Meanwhile, the acetylation of tau was also found in the brains of AD patients, which not only resulted in the disability of mitochondria fission by decreasing mitofusion proteins, but also impaired mitochondrial biogenesis via reducing the level of PGC-1α [166]. Collectively, these studies indicate that tau is intimately correlated with mitochondrial-dependent glucose metabolisms and insulin signaling in the brain.

Many theories of AD etiology have been devised, such as the amyloid cascade hypothesis [167], mitochondrial hypothesis [168], cholinergic hypothesis [169], neuroinflammatory hypothesis [170], oxidative stress hypothesis [171], insulin resistance hypothesis [172], and calcium hypothesis [173]. They are all supported by substantial clinical research and experimental data. In the current paper, we try to put together the data that correlate with insulin signal, Aβ overproduction, and tau phosphorylation to illustrate a chain of evidence for future pharmacological studies in this field.

As the evidence accumulated, we proposed that the insulin signaling pathway plays an important role in AD pathologies. Furthermore, the impairment of the substrate level of phosphorylation may be involved in hyperphosphorylated and truncated tau-induced mitochondrial damage. In the earlier stage of AD, anti-diabetes agents such as vanadium compounds were able to prevent or postpone the initiation of tau pathology by modulating the insulin signaling pathway. However, further studies are needed to investigate whether vanadium compounds have any protective function on the mitochondria.

**Author Contributions:** J.Y. and N.L. wrote the manuscript. Z.H. and G.Y. draw the figures. Q.L. and N.L. revised the manuscript. All authors reviewed and concurred the final manuscript. All authors have read and agreed to the published version of the manuscript.

**Funding:** This research was funded by Shenzhen-Hong Kong Institute of Brain Science-Shenzhen Fundamental Research Institutions (2021SHIBS0003).

**Acknowledgments:** Shenzhen-Hong Kong Institute of Brain Science-Shenzhen Fundamental Research Institutions.

**Conflicts of Interest:** The authors declare no conflict of interest.

**Abbreviations**

| | |
|---|---|
| Aβ | amyloid-beta |
| AD | Alzheimer's disease |
| ADAM10 | a disintegrin and metalloproteinase 10 |
| AGD | argyrophilic grain disease |
| AICD | APP intracellular domain |
| APOE | apolipoprotein E |
| APP | amyloid precursor protein |
| BEOV | bis(2-ethyl-3-hydroxy-4-pyronato) oxovanadium (IV) |
| BMOV | bis(maltolato) oxovanadium(IV) |
| CAA | cerebral amyloid angiopathy |
| CBD | corticobasal degeneration |
| CDK5 | cyclin-dependent kinase 5 |
| COX2 | cyclooxygenase-2 |
| CTE | chronic traumatic encephalopathy |
| CTF-83 | C-terminal fragment |
| CNS | central nervous system |
| Drp1 | dynamin-related GTPase |
| FAD | familial AD |
| FTDP-17 | frontotemporal dementia with Parkinsonism linked to chromosome 17 |
| GIP | glucose-dependent insulinotropic polypeptide |
| GLUT4 | glucose transporter 4 |
| GSIS | glucose-stimulated insulin secretion |
| GSK-3β | glycogen synthase kinase -3 |
| IDE | insulin degrading enzyme |
| IGF | insulin-like growth factor |
| iNOS | inducible nitric oxide synthase |
| IR | insulin receptor |
| IRS2 | insulin receptor substrate |
| LTP | long-term potentiation |
| MFN | mitofusion |
| NFTs | neurofibrillary tangles |
| Nrf2 | nuclear factor-erythroid-2-related factor 2 |
| PGC-1α | proliferator activated receptor gamma coactivator 1 |
| PI3K | phosphatidylinositol-3-kinase |
| PiD | Pick's disease |
| PP2A | protein phosphatase 2A |
| PPARγ | proliferator-activated receptor gamma |
| PS1/2 | presenilin-1/2 |
| PSP | progressive supranuclear palsy |
| PTEN | phosphatase and tension homologue on chromosome 10 |
| PTPase | protein tyrosine phosphatase |
| SCS | Succinyl-CoA synthetase |
| SCOT/OXCT1 | Succinyl-CoA3-ketoacid-CoA transferase catalyzes |
| STZ | streptozotocin |

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
