# Peer review of "The Deficits of Insulin Signal in Alzheimer’s Disease and the Mechanisms of Vanadium Compounds in Curing AD"

_cimb, doi:10.3390/cimb45080402_

Round 1

Reviewer 1 Report

Comments and Suggestions for Authors

MANUSCRIPT: 2538066

TITLE: The deficits of insulin signal in Alzheimer’s disease and the mechanisms of vanadium compounds in curing AD

The manuscript 2538066The deficits of insulin signal in Alzheimer’s disease and the mechanisms of vanadium compounds in curing AD”, presents a review of the literature.

The manuscript presented is well structured.

The review is clearly written, well systematized and comprehensive for the topic, and literature cited is adequate and most of the papers cited are from the last five years.

Similar reviews are not known and it is of interest to the scientific community.

The conclusions are consistent and in accordance with the quotes listed.

However, some questions remain to be clarified and solved and the manuscript in the current form must be revised in minor several points as follows comments:

1. References must be presented in accordance with the Reference List and Citations Style Guide for MDPI Journals. Please download the full MDPI Reference List and Citations Style Guide at MDPI | Reference List and Citations Style Guide and proceed as per the document and present the references as per the document and include the doi and/or ISBN.

2. List of references. The manuscript has 174 reference numbers and the list of references has only 145. Please present the complete list of references according to those cited in the text of the manuscript.

Other points

3. Line 31 - Please change “VO3-” by “VO3-”.

4. Line 32 - Please change “VO2+” by “VO2+”.

5. Line 61 - Please change “VOSO4” by “VOSO4”.

6. Line 31 - Please change “Ca2+” by “Ca2+”.

7. Line 192 - Please - Latin expressions should be written in italics, I would recommend writing the expression "et al" throughout the manuscript in italics.

8. Line 249 – MCI abbreviation. Please write in full the first time the authors use the abbreviation or put it in the list of abbreviations. It is recommended that authors review all text and complete the list of abbreviations.

9. Line 407 - Please change “va-nadium” by “vanadium”.

Author Response

Response to Reviewer

Reviewer 1

The manuscript 2538066 “The deficits of insulin signal in Alzheimer’s disease and the mechanisms of vanadium compounds in curing AD”, presents a review of the literature.

The manuscript presented is well structured.

The review is clearly written, well systematized and comprehensive for the topic, and literature cited is adequate and most of the papers cited are from the last five years.

Similar reviews are not known and it is of interest to the scientific community.

The conclusions are consistent and in accordance with the quotes listed.

However, some questions remain to be clarified and solved and the manuscript in the current form must be revised in minor several points as follows comments:

Response:We appreciated the comment very much, thanks to the reviewer for these kindly advices. 

  1. References must be presented in accordance with the Reference List and Citations Style Guide for MDPI Journals. Please download the full MDPI Reference List and Citations Style Guide at MDPI | Reference List and Citations Style Guide and proceed as per the document and present the references as per the document and include the doi and/or ISBN.

Response: We thank to reviewer for the kindly suggestion, the citation style was revised as MDPI reference.

  1. List of references. The manuscript has 174 reference numbers and the list of references has only 145. Please present the complete list of references according to those cited in the text of the manuscript.

Response: We thank to reviewer for this advice, the citations were updated.

Other points

  1. Line 31- Please change “VO3-” by “VO3-”.
  2. Line 32- Please change “VO2+” by “VO2+”.
  3. Line 61- Please change “VOSO4” by “VOSO4”.
  4. Line 31- Please change “Ca2+” by “Ca2+”.
  5. Line 192- Please - Latin expressions should be written in italics, I would recommend writing the expression "et al" throughout the manuscript in italics.
  6. Line 249– MCI abbreviation. Please write in full the first time the authors use the abbreviation or put it in the list of abbreviations. It is recommended that authors review all text and complete the list of abbreviations.
  7. Line 407 - Please change “va-nadium” by “vanadium”.

Response: We thank to reviewer for these advices, the typo had been corrected carefully at all.

Reviewer 2 Report

Comments and Suggestions for Authors

A paper submitted for review entitled. "The deficits of insulin signalling in Alzheimer's disease and the mechanisms of vanadium compounds in curing AD" written by Nan Li's team is a review article on the use of vanadium compounds in the treatment of Alzheimer's disease (AD). Given the seriousness of the problem of AD in modern society, such work is needed as it allows us to summarise our knowledge to date, which is useful for planning future research. The authors focused on vanadium compounds on the premise that AD can be treated as type-3 diabetes. It seems all the more likely that the review will find many readers. I recommend the paper for publication, but suggest the authors make a few corrections.

The paper should be rewritten in terms of chapters and subchapters. The introduction should be chapter number 3, and sequentially 3.1.Effect of vanadium on AZS curing; 3.2.Potential mechanisms of vanadium action in AZS curing.

The introduction should be rewritten. The last paragraph of the Introduction section should state: what type of review it is, what period it covers and what databases were searched for information.

Summary section "As evidence accumulated, we proposed that the insulin signalling pathway plays an important role in AD pathologies. " - as this is a review paper, this sentence should be changed to - The accumulated evidence supports that the insulin signalling pathway plays an important role in AD pathologies.

Minor mistakes:

Line 199-correct the sentence. It looks like something is missing.

Line 306 –the superscript is needed.

many split word errors such as 'hyperphosphory-lated' should be corrected

line 335 missing spaces like in here “PGC-1αwa”s - please correct .

line 336 unnecessary spaces - please correct 

Author Response

Response to Reviewer

Reviewer 2

A paper submitted for review entitled. "The deficits of insulin signalling in Alzheimer's disease and the mechanisms of vanadium compounds in curing AD" written by Nan Li's team is a review article on the use of vanadium compounds in the treatment of Alzheimer's disease (AD). Given the seriousness of the problem of AD in modern society, such work is needed as it allows us to summarise our knowledge to date, which is useful for planning future research. The authors focused on vanadium compounds on the premise that AD can be treated as type-3 diabetes. It seems all the more likely that the review will find many readers. I recommend the paper for publication, but suggest the authors make a few corrections.

Response:We are grateful of the kindly comments of the reviewer. Thanks very much!

The paper should be rewritten in terms of chapters and subchapters. The introduction should be chapter number 3, and sequentially 3.1. Effect of vanadium on AZS curing; 3.2.Potential mechanisms of vanadium action in AZS curing.

The introduction should be rewritten. The last paragraph of the Introduction section should state: what type of review it is, what period it covers and what databases were searched for information.

Response:Thanks for this helpful suggestion, the introduction and Chapter 3 had been rewritten. The data of this review are based on searching the coupled key words including APP and glucose/insulin, tau and glucose/insulin, Alzheimer’s disease and glucose/insulin, vanadium and Alzheimer’s disease on Pubmed.

Summary section "As evidence accumulated, we proposed that the insulin signalling pathway plays an important role in AD pathologies. " - as this is a review paper, this sentence should be changed to - The accumulated evidence supports that the insulin signalling pathway plays an important role in AD pathologies.

Response:Thanks for this kindly suggestion. This sentence had been revised.

Minor mistakes:

Line 199-correct the sentence. It looks like something is missing.

Line 306 –the superscript is needed.

many split word errors such as 'hyperphosphory-lated' should be corrected

line 335 missing spaces like in here “PGC-1αwa”s - please correct .

Response: We thank to reviewer for these advices, all these mistakes had been corrected carefully.
